# Comparative Analysis of Epicuticular Lipids in *Locusta migratoria* and *Calliptamus italicus*: A Possible Role in Susceptibility to Entomopathogenic Fungi

**DOI:** 10.3390/insects13080736

**Published:** 2022-08-16

**Authors:** Mariya D. Ganina, Maksim V. Tyurin, Ulzhalgas T. Zhumatayeva, Georgy R. Lednev, Sergey V. Morozov, Vadim Yu. Kryukov

**Affiliations:** 1N.N. Vorozhtsov Novosibirsk Institute of Organic Chemistry, Siberian Branch of Russian Academy of Sciences, Academician Lavrentyev Ave. 9, 630090 Novosibirsk, Russia; 2Institute of Systematics and Ecology of Animals, Siberian Branch of Russian Academy of Sciences, Frunze Str. 11, 630091 Novosibirsk, Russia; 3Department of Plant Protection and Quarantine, Faculty of Agrabiology, Kazakh National Agrarian Research University, Abai Avenue 8, Almaty 050010, Kazakhstan; 4All-Russian Institute of Plant Protection, Podbelskogo Avenue 3, St. Petersburg, 196608 Pushkin, Russia

**Keywords:** alkanes, insect cuticle, climatic adaptations, mycoses, *Metarhizium*

## Abstract

**Simple Summary:**

The surface lipids of insects protect them from desiccation and may modulate susceptibility to fungal infections. We conducted a comparative analysis of cuticular lipids of the migratory locust and Italian locust. The former inhabits relatively wet landscapes and the latter more arid ones. We analyzed cuticular lipids of these species by gas chromatography with mass spectrometry and found that the Italian locust has a hydrocarbon profile shifted toward long chains as well as a higher content of di- and trimethyl branched hydrocarbons, which is most likely an adaptation to the arid climate and strong temperature fluctuations in its habitats. Meanwhile, the surface of the Italian locust proved to be more hospitable for fungi. The number of *Metarhizium* conidia attached to the Italian locust cuticle was three-fold greater as compared to the migratory locust. Mortality due to the fungal infection was faster in the Italian locust under laboratory conditions. We propose that species inhabiting arid landscapes rarely encounter fungal pathogens and primarily deal with the problem of desiccation. Therefore, they can afford a cuticle that is hospitable to fungal pathogens.

**Abstract:**

Cuticular lipids protect insects from desiccation and may determine resistance to fungal pathogens. Nonetheless, the trade-off between these lipid functions is still poorly understood. The migratory locust *Locusta migratoria* and the Italian locust *Calliptamus italicus* have dissimilar hygrothermal preferences: *L. migratoria* inhabits areas near water bodies with a reed bed, and *C. italicus* exploits a wide range of habitats and prefers steppes and semideserts with the predominance of sagebrushes. This paper presents significant differences between these species’ nymphs in epicuticular lipid composition (according to gas chromatography with mass spectrometry) and in susceptibility to *Metarhizium robertsii* and *Beauveria bassiana*. The main differences in lipid composition are shifts to longer chain and branched hydrocarbons (di- and trimethylalkanes) in *C. italicus* compared to *L. migratoria*. *C. italicus* also has a slightly higher *n*-alkane content. Fatty acids showed low concentrations in the extracts, and *L. migratoria* has a wider range of fatty acids than *C. italicus* does. Susceptibility to *M. robertsii* and the number of conidia adhering to the cuticle proved to be significantly higher in *C. italicus*, although conidia germination percentages on epicuticular extracts did not differ between the species. We propose that the hydrocarbon composition of *C. italicus* may be an adaptation to a wide range of habitats including arid ones but may make the *C. italicus* cuticle more hospitable for fungi.

## 1. Introduction

Insect surface lipids perform a lot of physiologically important functions, the main ones being the prevention of water loss and inter- and intraspecific communication [1,2]. Moreover, cuticular lipids can contribute to the pathogenicity of microorganisms that penetrate through the cuticle [3,4,5]. The main epicuticular compounds are hydrocarbons and fatty acids, and their profiles are specific to insect taxa [1,2,6]. Cuticular lipid composition is significantly different from that of hemolymph and fat body, which contain mainly triglycerides and diglycerides [7]. Insect cuticular hydrocarbons are synthesized in oenocytes of the epidermal layer of integuments and are transported by lipophorin to epidermal cells via pore canals and epicuticular channels [6]. Both linear and branched hydrocarbons may be synthesized in different insect taxa, and the mechanisms of their synthesis have been reviewed by Blomquist [8]. Briefly, hydrocarbons are synthesized from long-chain fatty acetyl-coenzyme A by reductive decarboxylation. The elongation of the carbon chain proceeds via the addition of two carbon units by a malonyl-coenzyme A derivative, and the addition of methyl branches occurs by the insertion of propionate (a derivative of methylmalonyl-coenzyme A). The hydrocarbons can be oxidized to secondary alcohols by cytochrome P450 enzymes and oxygen (O_2_) and then can be oxidized to ketones and form wax esters [9].

It is known that cuticular hydrocarbon composition is dependent on climatic conditions of insect habitats [10,11,12]. From the physical standpoint, there is a lipid melting model suggested by Ramsay [13] explaining the role of hydrocarbons in protection from desiccation [14]. According to the model, surface lipids stay in a solid state at optimal temperatures thereby providing a strong barrier to transpiration. As temperature increases, lipids begin to melt resulting in higher cuticular permeability and an increase in desiccation. Temperature and properties of melting depend on hydrocarbon composition. Linear saturated hydrocarbons (*n*-alkanes) are known to have higher melting points as compared to methyl-branched alkanes and unsaturated hydrocarbons (alkenes) [15,16]. Menzel and coworkers [10] suggested that hydrocarbon composition should reflect the trade-off between the epicuticle functions. On the one hand, the epicuticle should be viscous to prevent desiccation. This property is determined by the proportion of *n*-alkanes and/or the average chain length in the hydrocarbon profile. On the other hand, the epicuticle should possess fluidity for effective signaling and for a uniform distribution of epicuticular compounds on the insect surface. The same point of view was expressed by Lockey and Oraha [17] and by Gibbs [14]. Similarly, Chung and Carroll [18] believe that *n*-alkanes are unlikely to act as semiochemicals owing to poor structural diversity. Hydrocarbons involved in chemical communication are much more diverse in chemical structure (e.g., alkenes and dienes) and have a high capacity for information but a weak potential for waterproofing because of a relatively low melting point. Hydrocarbons with an intermediate melting point, for example, methyl-branched alkanes as well as alkenes, can have dual traits, i.e., may influence both desiccation resistance and chemical communication.

There are studies showing that the species occupying arid habitats have higher melting points of cuticular hydrocarbons as compared to species from humid habitats [10,11,19]. Moreover, different populations within a species may differ in these properties. For instance, populations of the grasshopper *Melanoplus sanguinipes* (Fabr.) from southern latitudes of Northern America are characterized by a higher melting point of cuticular lipids as compared to those from northern latitudes [20]. Epicuticular hydrocarbon composition in xerophilous species may be shifted toward the prevalence of *n*-alkanes [11,19]. In addition to hydrocarbons, an important role in waterproofing can be played by fatty acids. Lockey and Oraha [17] supposed that free fatty acids are located on the border of the cuticle surface and lipid layer to fix nonpolar lipids on the polar cuticle. 

Among locusts, there are species adapted to both dry and relatively moist habitats. Particularly, some economically important agricultural pests—the migratory locust *Locusta migratoria* (L.) and the Italian locust *Calliptamus italicus* (L.)—have different habitat preferences. The migratory locust prefers to nest along a riverbank, lake shore, or sea coast or across swampy meadows with a reed bed [21]. The Italian locust exploits a wide range of habitats and prefers more arid landscapes such as steppes and semideserts with a mosaic grass cover and predominance of sagebrushes *Artemisia* spp. [22]. These two species of locusts could be suitable for elucidating the participation of cuticular lipids in adaptation to different habitats as well as the role of lipid composition in susceptibility to fungal infections. 

There are studies describing cuticular lipids of adult *L. migratoria* ssp. *migratorioides* [17,23,24], but there are no studies about cuticular lipids of *C. italicus*. Nevertheless, there are research articles describing cuticular lipids of *S. gregaria*, which inhabits subtropical dry regions. Lockey and Oraha [17,24] conducted a comparative analysis of the cuticular lipid composition of *L. migratoria* and *S. gregaria*; the main dissimilarity was a greater *n*-alkane amount and the presence of trimethylalkanes in *S. gregaria*. Those authors proposed that these properties of the *S. gregaria* lipid layer may represent an adaptation to arid habitats because a large *n*-alkane amount decreases desiccation. 

Ascomycetes *Metarhizium* and *Beauveria* are natural pathogens of locusts and grasshoppers, and products of these pathogens are used for biological control of locust populations [25]. As pointed out above, various cuticular compounds play an important part in insect susceptibility to insect pathogenic fungi infecting hosts through the integument [26]. For example, insects with saturated and branched hydrocarbons (alkanes) may be more susceptible to fungi than are insects with unsaturated chains (alkenes and alkadienes) [3]. The lipid amount and profile determine the hydrophobicity of the cuticle, which is an important factor for conidia attachment to the insect surface [27]. Epicuticular extracts with the predominance of methyl-branched alkanes as well as authentic linear alkanes stimulate the germination of fungal conidia [28,29]. It is known that linear and methyl-branched hydrocarbons are utilized by *Metarhizium* and *Beauveria* [30] via the cytochrome P450 enzyme system, and alkanes are transformed into alcohols, aldehydes, and fatty acids [4,31]. Jarrold and coworkers showed that lipid compounds of the *S. gregaria* cuticle affect the differentiation of *Metarhizium* infection structures [28]. Those authors hypothesized that polar compounds (free and esterified fatty acids, glucose, and amino acids) stimulate fungal germination before fungi can take up nonpolar components. Hydrocarbon composition and fatty acid composition of the insect cuticle and of internal tissues significantly change during the development of a fungal infection; however, these effects are not straightforward and depend on insect and pathogen species [32,33,34,35]. Probably, the alterations are caused not only by the direct utilization of lipids by fungi but also by changes in host metabolic pathways during fungal infections. Notably, a number of fatty acids (especially with short chains) can inhibit germination and mycelial growth of entomopathogenic fungi in vitro [36,37] and can influence the activity of virulence-related enzymes [38,39]. 

To the best of our knowledge, there are no research papers describing the impact of epicuticular lipids of different locust species on their susceptibility to entomopathogenic fungi. We demonstrated previously that under identical hygrothermal conditions, *Calliptamus* species are more susceptible than *L. migratoria* to fungi *Metarhizium* and *Beauveria* [40]. Nonetheless, it is not known whether the differences in infection susceptibility are linked with cuticular lipid composition. In this work, we performed a comparative analysis of epicuticular compounds of *L. migratoria* and *C. italicus* nymphs—hydrocarbons and fatty acids—by gas chromatography coupled with mass spectrometry (GC-MS). In addition, (i) the number of *Metarhizium robertsii* conidia adhering to the cuticle of both species, (ii) the conidia germination level (%) on cuticular extracts, and (iii) mortality dynamics in response to the fungal infections were determined.

## 2. Methods

### 2.1. Insects and Fungi

Third-instar locust nymphs were collected from natural populations in Southeast Kazakhstan (Almaty region) in May 2019 (collection sites: *L. migratoria* ssp. *migratoria*, 45.385162° N 75.249230° E; *C. italicus*, 43.518412° N 76.830550° E). Experimentation on nymphs allows us to accurately link their origin to certain habitats, in contrast to adults, who can migrate substantially. The nymphs were collected with an entomological hoop net into cages with metal mesh 80 × 80 cm and transported to the laboratory within 1 day. A mixture of reeds and sedges was provided as food for *L. migratoria,* and a mixture of *Artemisia* species for *C. italicus*.

Entomopathogenic fungus *M. robertsii* isolate P-72 (GenBank accession No. KP172147.2) and *Beauveria bassiana* isolate UK-4 from the collection of microorganisms at the Institute of Systematics and Ecology of Animals (the Siberian Branch of the Russian Academy of Sciences; SB RAS) were used for bioassays. Fungal conidia were cultivated on twice-autoclaved millet, then dried for 10 d at 24 °C, and sifted through a soil sieve. The conidia were suspended in sterile 0.03% (*v*/*v*) aqueous Tween 20 for insect inoculation. Concentrations of conidia were determined by means of a Neubauer hemocytometer. The conidia germination level on Sabouraud dextrose agar was >95%.

### 2.2. Cuticular Lipid Extraction and Derivatization

The insects were placed into 3 L glass jars with filter paper (80 insects per jar) and euthanized with chloroform for 5 min to minimize contamination of extracts with a regurgitate and feces. Surface lipids were extracted with hexane:diethyl ether mixture (2:1, *v*/*v*), namely, the insects were put in 100 mL glass jars (80 insects per jar), 52 mL of the solvent mixture was added, and the jars were agitated at 160 rpm for 5 min at room temperature. The extracts were transferred into porcelain cups for evaporation in a laminar flow cabinet for 3 h at 25 °C. Next, the extracts were placed into 10 mL vials, evaporated overnight at 25 °C, and stored at −20 °C till the analysis. Six biological replicates were processed and analyzed for each species (one replicate = 80 nymphs). Before GC-MS, the extracts were weighed and derivatized with methanol in sulfuric acid to assay total fatty acids [41].

### 2.3. GC-MS Analysis

A 6890 N gas chromatograph with mass-selective detector 5975 N (Agilent Technologies, Santa Clara, CA, USA) was used for chemical identification and quantification of hydrocarbons and fatty acids. Quartz capillary column HP-5MS (5% biphenyl- and 95% dimethyl silicone, 30 m × 0.25 mm, film thickness 0.25 μm) was utilized. The volume of the injected sample was 1 μL in splitless mode. Helium served as a carrier gas, and the flow rate was 0.8 mL/min. The temperature of the injector and interface was 310 °C, and the temperature of the column was programmed to go up from 50 to 310 °C at 2 °C/min and was held at 310 °C for 13 min (regime Sterol2G310). The mass spectra were recorded via electron impact ionization at ionization energy of 70 eV. *n*-Tetracosane was used as an internal standard; the quantification was performed with external standards: *n*-dotriacontane for hydrocarbons and methyl stearate for fatty acids. The quantification was applied to total ion current chromatogram peaks with area > 0.5%. To identify methyl-branched hydrocarbons, we employed the algorithm proposed by Nelson et al. [42], with minor modifications. Determination of structure was conducted according to three criteria: (1) a comparison of calculated linear retention indexes (LRI) with those described in the literature for methyl-branched hydrocarbons of locusts and other insects [17,23,42,43]; (2) characteristic ions in the mass spectra formed during carbon chain cleavage at a methyl branch point were compared with the decay of a proposed structure; (3) methyl-branched hydrocarbon biosynthesis pathways imposing restrictions on the structure of the molecule were taken into account [2,42]. A detailed description of detection of characteristic ions is given in Appendix A. The method of chromatogram reconstruction according to characteristic ions was applied to the analysis of complicated peaks (examples for several peaks are shown in Appendix A). Fatty acid identification was carried out with the help of the NIST 02 MS database included in Agilent G 170 1 AA ChemStation software. In cases of overlap between hydrocarbon and fatty acid peaks, the method of reconstruction of a chromatogram according to characteristic ions was used. The amounts of hydrocarbons and fatty acids were calculated in μg/mg of an extract and also recalculated in milligrams per gram of insect body weight.

### 2.4. Susceptibility to Fungal Infections

For a survival assay, *M. robertsii* and *B. bassiana* at a concentration of 10^7^ conidia/mL were used. Insects were dipped for 10 s either in the fungal suspension in an aqueous Tween 20 solution (0.03%) or in an aqueous Tween 20 solution free from conidia (control). After this inoculation, the insects were maintained in 1 L ventilated plastic containers (ten nymphs per container) at 24 °C under the 16:8 h photoperiod (light:dark). Feeding and mortality estimates were made daily for 12 d. The bioassay was performed on four biological replicates (one replicate = 10 nymphs); the whole experiment was independently conducted three times.

### 2.5. Adhesion and Germination Assays

Modified techniques of Ment and coworkers [37,44] were used for *Metarhizium* adhesion and germination assays. Nymphs were infected by dipping in an *M. robertsii* suspension (10^8^ conidia/mL) for 10 s and maintained as described above. Six hours postinoculation, the insects were vortexed in an aqueous Tween 20 solution (0.05%) for 1 min at 500 rpm to remove unadhered conidia. Then, four nymphs were placed into a 10 mL vial with 5 mL of dichloromethane and agitated for 5 min at 200 rpm. The insects were removed from the vials, and dichloromethane was evaporated to dryness. After that, the conidial sediments were resuspended in 2 mL of an aqueous Tween 20 (0.05%) solution at 3000 rpm for 5 min, and the conidia concentration was determined on a hemocytometer and calculated per nymph. Eight biological replicates (one replicate = four nymphs) for both species were used for the analysis.

To determine the effect of the extracts on the conidia germination level, the extracts (hexane:diethyl ether at 2:1, *v*/*v*) were placed dropwise on the surface of agarose in Petri dishes [26] to obtain a final concentration: an extract from one nymph per 2 cm^2^ plug of agarose. Control plugs were treated with the pure solvent mixture. Next, an *M. robertsii* conidia suspension in an aqueous Tween 20 solution (0.03%) was spread on the plugs (20 μL, 10^6^ conidia/mL) and dried under laminar air flow for 20 min. The plugs were incubated at 26 °C for 16 h, and then the numbers of germinated and nongerminated conidia were calculated with the help of light microscopy. Three biological replicates (10 fields of view in each) per treatment group were used.

### 2.6. Statistics

The data were checked for normality by the Shapiro–Wilk *W* test. Significance of differences between the two species in terms of individual compounds, compound classes, adhesion, and germination levels was determined by the *t*-test in case of the normal distribution and by the Mann–Whitney *U* test for non-normal distribution. Differences in mortality dynamics were estimated by the log-rank test. Additionally, Abbott’s correction followed by the χ^2^ test was applied to each day of the bioassay. The analyses were performed in PAST 3.0 [45] and SigmaStat 3.1 (Systat Software Inc., Tulsa, OK, USA).

## 3. Results

### 3.1. Cuticular Lipids

Epicuticular hydrocarbons, fatty acids, and other lipids of *C. italicus* and *L. migratoria* were analyzed to understand their possible involvement in adaptation to different habitats and in susceptibility to fungal infections. Relative dry weights of *C. italicus* and *L. migratoria* epicuticular extracts were similar: 0.62 ± 0.02 and 0.61 ± 0.01 mg/(g of insects), respectively (t = 0.52, df = 10, *p* = 0.62, Appendix A). Saturated hydrocarbons (78–79% of an extract), saturated and unsaturated fatty acids (1.5–2.7% of an extract), and ketones (up to 5% of an extract) were detectable in the extracts. The identified compounds and their contents per unit weight of an extract or of insects are given in Appendix A.

#### 3.1.1. Hydrocarbons

Estimation of hydrocarbon amounts in μg/(mg of an extract) showed that the total hydrocarbon amount does not differ significantly between *C. italicus* and *L. migratoria* (t < 0.14, df = 10, *p* > 0.8). Hydrocarbon composition differs between the species (Figure 1 and Figure 2, see also Appendix A). The hydrocarbon profile of *C. italicus* was found to be shifted toward longer chains as compared to *L. migratoria* (Figure 2A,B). In particular, the amounts of relatively short-chain hydrocarbons (C25–C29) are similar between the species. The amount of medium-chain hydrocarbons (C30–C34) is 5.9-fold lower in *C. italicus* than in *L. migratoria* (t = 10.2, df = 10, *p* < 0.0001). In contrast, the long-chain hydrocarbon amount (C35–C39) is 1.9-fold greater in *C. italicus* (t = 5.1, df = 10, *p* < 0.001). In addition, long-chain hydrocarbons C40 and C41 were registered in trace amounts in the extract of *C. italicus* but were not detectable in the *L. migratoria* extract (Appendix A). 

The hydrocarbon profiles of the two species were found to differ in the numbers of methyl branches on the carbon backbone (Figure 2C). Qualitatively, *C. italicus* contains a narrower spectrum of monoalkanes but a wider spectrum of di- and trimethylalkanes as compared to *L. migratoria*. Quantitatively, the *n*-alkane amount of *C. italicus* is 1.2-fold greater, but the difference is not significant (t = 1.45, df = 10, *p* = 0.18). Nevertheless, evaluation of the *n*-alkane proportion within the hydrocarbon class alone revealed a significantly greater level of *n*-alkanes in *C. italicus* (Z = 2.6, df = 10, *p* = 0.008). The major *n*-alkane in both species turned out to be *n*-C29, and its amount does not differ significantly between the species (Figure 2A). Another abundant hydrocarbon is *n*-C27, and its amount is two-fold greater in *C. italicus* than in *L. migratoria* (t = 6.3, df = 10, *p* < 0.001). 

The monoalkane amount is 7.2-fold less in *C. italicus* compared to *L. migratoria* (t = 10.1, df = 10, *p* < 0.0001, Figure 2C). The major monoalkanes of *L. migratoria* proved to be isomers with carbon numbers C30 and C32 (chain lengths of C29 and C31), which are present in *C. italicus* in trace amounts (Figure 2A). The major monoalkanes of *C. italicus* are isomers with a carbon number of C36 (chain length of C35), and their level is 1.8-fold greater as compared to *L. migratoria* (z = 2.5, df = 10, *p* = 0.013). 

The dimethylalkane amount is 1.6-fold greater in *C. italicus* than in *L. migratoria* (t = 4.0, df = 10, *p* = 0.003, Figure 2C). In both species, the major dimethylalkanes have a carbon number of C37 (chain length of C35), and their amount is 1.8-fold higher in *C. italicus* than in *L. migratoria* (t = 5.6, df = 10, *p* < 0.001, Figure 2A). Moreover, trimethylalkanes were identified in the extract of *C. italicus* but were absent in the *L. migratoria* extract. 

Notably, similar differences between *L. migratoria* and *C. italicus* in proportions of various hydrocarbon families were documented in terms of μg/(g of insects) (Appendix A).

#### 3.1.2. Fatty Acids

The fatty acid spectrum in the *C. italicus* epicuticle is narrower relative to *L. migratoria* (Figure 3, see also Appendix A). In particular, fatty acids C14–C34 were identified in the epicuticular extract of *L. migratoria*, where C16:0, C18:3, C28:0, and C30:0 turned out to be the most abundant. Fatty acids C16–C20 were found in the extracts of *C. italicus*, and C16:0 is the most abundant. The total fatty acid amount is 14.1 ± 1.7 μg/(mg of the extract) in *C. italicus* and 21.1 ± 2.49 μg/(mg of the extract) in *L. migratoria* (t = 2.3, df = 10, *p* = 0.041). A similar pattern was observed in terms μg/(g of insects): 8.62 ± 0.93 μg/g in *C. italicus* and 12.9 ± 1.5 μg/g in *L. migratoria* (t = 2.4, df = 10, *p* = 0.036). The ratio of unsaturated fatty acids to the total lipid amount is 0.63% ± 0.10% in *L. migratoria* and 0.81% ± 0.07% in *C. italicus*, and the difference is not significant (t = 1.45, df = 10, *p* = 0.18).

#### 3.1.3. Other Lipids

Long-chain ketones belonging to homologous series C34–C40 were detected in the extracts of *C. italicus*. Their total amount was 50.8 ± 3.6 μg/(mg of the extract) (5% of extract weight). Mass chromatograms, mass spectra, and identified structures are presented in Appendix A. Ketones were not detectable in the extracts of *L. migratoria*. 

### 3.2. Susceptibility to Fungal Infections

A comparative analysis of the mortality dynamics of *C. italicus* and *L. migratoria* nymphs was performed after treatment with *M. robertsii* and *B. bassiana* conidia. Nymphs of *C. italicus* were more susceptible to *M. robertsii* than *L. migratoria* nymphs were (Figure 4A). Median lethal time (LT_50_) after treatment with the conidia was 12 ± 0.06 d for *L. migratoria* and 9 ± 0.06 d for *C. italicus* (log-rank test: χ^2^ = 55.2, df = 1, *p* < 0.0001). The mortality of the infected nymphs significantly differed from corresponding controls (χ^2^ > 38.4, df = 1, *p* < 0.0001). Differences in mortality between uninfected *L. migratoria* and uninfected *C. italicus* nymphs were insignificant (χ^2^ = 1.1, df = 1, *p* = 0.30), and the mortality rates were 13% and 21%, respectively, on day 12 post-treatment. Estimation of differences after Abbott’s correction also showed significant inequality in the mortality rate between *C. italicus* and *L. migratoria* nymphs after *M. robertsii* infection (days 8, 9, 10, and 11, χ^2^ > 6.0, df = 1, *p* < 0.014). The experiment was conducted independently three times, and the results were consistent.

A similar effect was registered after treatment of the nymphs with *B. bassiana* conidia (Figure 4B). Mortality due to the infection was faster in the *C. italicus* compared to *L. migratoria,* reaching 100% and 90%, respectively, on day 12 (log-rank test: χ^2^ = 6.8, df = 1, *p* = 0.009). Differences in mortality among untreated controls were insignificant: reaching 3% in *C. italicus* and 10% in *L. migratoria*. 

### 3.3. The Adhesion and Germination Assays

The prepenetration stages of fungal development on the cuticle (adhesion and germination) are dependent on the host’s lipid composition, which may determine the susceptibility of insects to fungal pathogens [3,27,28]. Therefore, we estimated the number of *M. robertsii* conidia adhering to the cuticle of both locust species in vivo and the rate of conidia germination in vitro. The number of *M. robertsii* conidia adhered to the cuticle of *C. italicus* nymphs was 3.3-fold greater as compared to *L. migratoria,* and the difference was significant (t = 3.7, df = 14, *p* = 0.002, Figure 5A). Epicuticular extracts of both species *C. italicus* and *L. migratoria* stimulated conidial germination of *M. robertsii* on agarose (7.5–8.0-fold compared to pure-solvent treatment, t > 5.4, df = 4, *p* < 0.006, Figure 5B). There were no significant differences in conidia germination between the extracts from *C. italicus* and *L. migratoria* (t = 0.38, df = 4, *p* = 0.73).

## 4. Discussion

This study points to significant differences in cuticular lipid composition between *L. migratoria* and *C. italicus* nymphs. The main dissimilarities are shifts to longer chain and methyl-branched hydrocarbons in *C. italicus* compared to *L. migratoria*. The differences in lipid composition were found to correlate with higher susceptibility of *C. italicus* to insect-pathogenic fungi as compared to *L. migratoria* and with a greater number of conidia adhering to the *C. italicus* cuticle. Below, we discuss the possible role of lipid profiles of these locusts in adaptation to different habitats as well as a probable contribution of these lipids to the susceptibility to fungal infection. 

We established that *C. italicus* contains a larger amount of long-chain alkanes C35–C39 relative to *L. migratoria*. In addition, *C. italicus* has a lower amount of monomethylalkanes but a larger amount of dimethylalkanes and trimethylalkanes than *L. migratoria* does. Furthermore, the proportion (%) of *n*-alkanes is slightly higher in *C. italicus*. We can hypothesize that these differences are caused by adaptations to dissimilar hygrothermal regimes. It is known that the melting point of hydrocarbons decreases in the following order: *n*-alkanes—monomethylalkenes—dimethylalkanes—alkenes, but molecular weight affects the melting point less strongly [15,19]. Regarding molecular structure, *n*-alkanes form tighter crystal packing, which provides a robust barrier to evaporation [17]. Methyl group insertion disrupts the ordering of the packing and raises fluidity [46]. Nevertheless, the cuticular lipid set represents a complicated multicomponent mixture that melts not abruptly but rather gradually across a wide range of temperatures [15]. Lockey and Oraha [17] suggested that waterproofing is provided mainly by *n*-alkanes, whereas methyl-branched compounds ensure hydrocarbon matrix fluidity, which is important for lipid secretion through pore channels, for a uniform distribution of polar components in the lipid matrix on the insect surface, and for the maintenance of the stability of the lipid layer, particularly during daily changes in temperature. Thus, the greater amount of di- and trimethylalkanes in *C. italicus* is likely to be an adaptation to wide temperature fluctuations in arid habitats. 

There are research articles showing correlations between cuticular lipid profiles and habitats of insects. By means of genera *Crematogaster* (Formicidae: Myrmicinae) and *Camponotus* (Formicidae: Formicinae) from different biogeographical environments as an example, Menzel and coauthors [10] demonstrated that cuticular lipid composition depends on precipitation load in ants’ habitats. Their species residing in humid landscapes have a lower proportion (%) of dimethylalkanes but a greater proportion of alkenes. The percentage of *n*-alkanes slightly goes up with increasing temperature. Both results are consistent with our data. At the same time, those authors showed that medium chain length does not depend on habitat conditions. Hadley and Schultz [11] analyzed differences in cuticular hydrocarbon profiles among tiger beetles (genus *Cicindela*) from dissimilar microhabitats. *Cicindela oregona* (dwelling along water bodies) was found to possess equal percentages of saturated and unsaturated hydrocarbons. *C. obsolete* (preferring dry grasslands) contains saturated hydrocarbons only (methyl-branched alkanes and *n*-alkanes), with a 1.5-fold greater amount of methyl-branched alkanes compared to the amount of *n*-alkanes. Moreover, in that study, *C. obsolete* manifested a greater percentage of long-chain hydrocarbons and a 1.5-fold larger total amount of hydrocarbons per unit of surface (cm^2^) as compared to *C. oregona*. The more xerophilous species *C. obsolete* loses water two-fold less than *C. oregona* does. Differences in molecular weight were observed among *Drosophila* species from different habitats [19]. The species *D. mojavensis* occurring in the Sonoran Desert predominantly contains hydrocarbons C29–C39, whereas cuticular lipids of *D. melanogaster* and other mesophilic species mainly consist of compounds with less than 30 carbon atoms. McClain and coworkers [12] investigated surface lipids in adults of Tenebrionid beetles from three climatic zones in the Namib desert. Those authors revealed that the absolute amount of surface lipids is greater in arid areas, but hydrocarbon composition differs insignificantly. It is likely that various insect taxa have distinct mechanisms underlying the formation of the epicuticular lipid layer for effective waterproofing. Nonetheless, general trends may be noticeable, such as upregulation of linear and methyl-branched alkanes and greater hydrocarbon chain length in insects inhabiting arid ecosystems. 

Regarding locusts, Lockey and Oraha [17] performed a comparative analysis of cuticular lipids in adults of the migratory locust *L. migratoria migratoriodes* and of the desert locust *S. gregaria*; the hydrocarbon spectrum of *S. gregaria* proved to be narrower as compared to *L. migratoria*, and molecular-weight ranges are equal between the two species. As for structure, the *n*-alkane amount in *S. gregaria* is higher, but the monomethylalkane amount (particularly 3-methylalkane) is lower relative to *L. migratoria*. The dimethylalkane amount does not differ considerably between the two species. In that work, trimethylalkanes were identified in *S. gregaria* (1.5%) but not in *L. migratoria*. Those authors [17] proposed that the large *n*-alkane amount in *S. gregaria* serves as an adaptation to the more arid climate, and di- and trimethylalkanes in the *S. gregaria* epicuticle expand the melting point range and provide stability to the lipid matrix during wide daily fluctuations of temperature in arid habitats. Nonetheless, from the ecological point of view, *S. gregaria* exploits temporary humid habitats in a milieu that is dry overall [47]. It has been shown experimentally that the optimal humidity range is similar between *S. gregaria* and *L. migratoria migratoriodes* (60–70%) [48,49]. Moreover, the ranges of temperature and humidity in which *S. gregaria* hoppers can develop are narrower as compared to *L. migratoria migratoriodes* hoppers. In particular, *L. migratoria migratoriodes* hoppers are better adapted to high humidity (80%) [48]. Therefore, the question of the correlation between cuticular hydrocarbon composition and adaptations to habitats within this pair of species remains open. To our knowledge, there are no comparative studies on thermo- and hygropreferences of *C. italicus* and *L. migratoria migratoria.* Nevertheless, it is known that in the studied region, the Italian locust exploits a wider range of habitats and prefers semideserts and dry steppes at different altitudes, whereas the migratory locust shows narrower ecological requirements and prefers reed beds along water bodies [21,22,50,51]. It is possible that the shifts to dimethylalkanes and trimethylalkanes in *C. italicus* (as compared to *L. migratoria*) are an adaptation to a wide range of hygrothermal conditions and/or their fluctuations in arid landscapes. We can also theorize that the longer hydrocarbon chains in *C. italicus* could compensate for the high proportion of branched hydrocarbons to improve viscosity and the melting point, thereby offering greater desiccation resistance.

It should be noted that there is a link between the number of methyl branches and the molecular weight of alkanes. Both species here possess predominantly *n*-alkanes in the near region of the profile (C25–C29), predominantly monomethylalkanes in the middle region of the profile (C30–C34), and dimethylalkanes and trimethylalkanes (in *C. italicus*) in the far region of the profile (C35–C39). The same trend has been documented in reports on *L. migratoria* and *S. gregaria* [17,23,24,52]. Molecular mass and the branching degree do not appear to be independent parameters. Menzel and coworkers [10] believe that the simultaneous presence of various hydrocarbon types is unlikely to be accidental if they originate from the same or distinct biochemical pathways. On the other hand, there may also be a restriction on the simultaneous output of different compound types. Chung and Carroll [18] state that a change in the production of one type of cuticular lipids may result in an alteration in the production of another type of cuticular lipids, which are involved in chemical communication or waterproofing, because insect cuticular lipids are synthesized by the common biochemical pathways involving acetyl-coenzyme A. Probably, a change in one cuticular hydrocarbon can have an indirect effect on other cuticular hydrocarbons or their types and on the dual trait of these lipids. There is likely to be a common biochemical mechanism regulating chain length and the methyl branching degree, and in the species under study, it works in opposite ways: *C. italicus* has a large amount of long-chain dimethylalkanes, whereas *L. migratoria migratoria* a large amount of medium-chain monomethylalkanes. Notably, the hydrocarbon profile of *L. migratoria migratoria* nymphs established in the present paper is very similar to the profile of *L. migratoria migratoriodes* adults described by Lockey and Oraha [17]. Despite differences in developmental stages and subspecies, the same chain lengths for different classes of alkanes (*n*-alkanes, monomethylalkanes, and dimethylalkanes) are predominant, and their relative abundance levels are similar too.

Regarding fatty acids, we found the opposite tendency. The spectrum of *L. migratoria* is shifted toward a longer chain region. In particular, very-long-chain fatty acids C22–C34 were registered along with medium-chain fatty acids C14–C20. *C. italicus* possesses only C16–C20 fatty acids. This finding is also consistent with other studies on adult *L. migratoria* and *S. gregaria*: very-long-chain fatty acids C28–C30 are dominant among polar compounds in *L. migratoria* wings, whereas C14:0, C16:0, and C18:0 predominate in *S. gregaria* [53,54]. From the standpoint of desiccation prevention, various fatty acids show patterns similar to those observed for hydrocarbons in terms of the unsaturation degree [17]. Notably, we detected only a low proportion of fatty acids in the epicuticular extracts of *L. migratoria* and *C. italicus* nymphs (1.5–2.7%); therefore, they unlikely play a leading part in desiccation resistance or susceptibility to infection.

Aside from the lipid melting point, another important parameter for waterproofing is the hydrophobicity of the lipid layer. An octanol–water partition coefficient (log *Kow*) is employed as an indicator of the hydrophobicity of certain compounds. Among short-chain *n*-alkanes, log *Kow* increases with the extension of carbon chain length but decreases from linear to branched alkanes [55]. It is important to mention that chain length has a stronger positive effect on hydrophobicity than the negative effect of the branching degree [55]. Due to the observed shift to long-chain hydrocarbons in the *C. italicus*, we can propose that its cuticle is more hydrophobic than that of *L. migratoria*, also in agreement with the different habitats of the species.

We registered a greater number of *M. robertsii* conidia adhering to the cuticle of *C. italicus* as compared to *L. migratoria*. It is well known that hydrophobic interactions perform the key function in passive nonspecific conidia adhesion to an insect epicuticle [56]. It is possible that longer chain hydrocarbons in the *C. italicus* cuticle ensure high hydrophobicity and potent conidia adhesion. Further investigation may be focused on adhesion forces between conidia and cuticles of mesophilic and xerophilous locusts using atomic force microscopy (e.g., [57]). These data can help to understand pathogeneses of fungal infections in insects adapted to various hygrothermal conditions. 

We demonstrated that cuticular extracts of both *L. migratoria* and *C. italicus* stimulate the germination of *M. robertsii* conidia on agarose. It has been shown previously that authentic hydrocarbons or extracts containing predominantly hydrocarbons promote the germination of various *Metarhizium* species [28,29,58]. It is noteworthy that we did not observe significant differences in conidia germination between the extracts from the two tested locusts, implying that adhesion but not germination makes the key contribution to the higher susceptibility of *C. italicus* to *M. robertsii*. 

It is important that Levchenko et al. [59] found *S. gregaria* (closer in cuticular lipid composition to *C. italicus*) to be also more susceptible to entomopathogenic ascomycetes (*M. robertsii*, *B. bassiana*, and *B. brongniartii*) than *L. migratoria* was; however, conidia adhesion was not evaluated there. It is noteworthy that xerophilous species *Mioscirtus wagneri* (Ev.) and *Calliptamus barbarus* (Costa) and the Dociostaruini species complex are more susceptible to *Metarhizium* and *Beauveria* fungi than are more mesophilic *L. migratoria* and *Mecostethus alliaceus* (Germ.) in laboratory experiments under the same temperature and humidity conditions [40]. Similar effects were obtained by Kassa [60] in field cage experiments on *L. migratoria* and *Cryptocatantops haemorrhoidalis* (Krauss) after treatment with either areal conidia or submerged spores of *M. acridum*. Physiological, evolutionary, and ecological mechanisms of high susceptibility to fungal infections in xerophilous species are unclear. We can speculate that because dry and hot habitats are unfavorable for fungi, xerophilous locusts rarely encounter fungal infections and consequently do not require highly developed antifungal defenses [40,59]. Under the conditions of a sparse grass cover and/or high altitudes, it is difficult for fungi to survive and develop on the cuticle of terrestrial insect stages owing to potent UV radiation and low humidity. Therefore, the main strategy of xerophilous species is protection against desiccation, not against fungal pathogens, and they can afford the cuticle hospitable for fungi. For confirmation of the selection processes underlying the formation of resistance to fungi among mesophilic and xerophilous locusts, studies on cellular and humoral immune responses to fungal infections are required. In addition, a comparative study on the fungal load and locust susceptibility to infections in natural habitats of the investigated region may be promising.

Notably, the methods used in this work have limitations because we analyzed lipid composition, adhesion, and germination levels for the whole insect surface. On the other hand, lipid composition may vary among insect body parts [17,61]. Adhesion and germination levels may also vary among organs of arthropods [27,62]. Therefore, a future investigation may also address lipid composition, fungal adhesion, and the differentiation of infection structures among certain organs of locusts.

## 5. Conclusions

This is the first comparative analysis of cuticular lipids of *L. migratoria* and *C. italicus* nymphs. Linear and methyl-branched alkanes are predominant in the lipid profiles of both species, although the lipid profiles are considerably different between these species: (1) *C. italicus* contains a larger amount of long-chain hydrocarbons (C35–C39) as compared to *L.*
*migratoria*; (2) *C. italicus* possesses a lower amount of monomethylalkanes but a larger amount of di- and trimethylalkanes relative to *L. migratoria*; (3) *n*-alkane proportion is slightly higher in *C. italicus*, and (4) *C. italicus* has a narrower spectrum of fatty acids than *L. migratoria* does. Higher susceptibility to fungal infections caused by *M. robertsii* and *B. bassiana* was registered for *C. italicus* compared to *L. migratoria*. A greater number of *M.*
*robertsii* conidia adhering to the cuticle was observed, which correlated with an elevated amount of long-chain hydrocarbons in *C. italicus*. We believe that epicuticular hydrocarbons of *C. italicus* may be an adaptation to a wide range of habitats (including arid steppes and semideserts) but may make the *C. italicus* cuticle more hospitable for fungi. Because a propensity for higher susceptibility to fungi has been documented for other xerophilous locusts compared to mesophilic ones in laboratory assays, future research may be devoted to an immune response to fungi in locusts with distinct hygrothermal preferences as well as to the development of fungal infections in natural habitats. 

## Figures and Tables

**Figure 1 insects-13-00736-f001:**
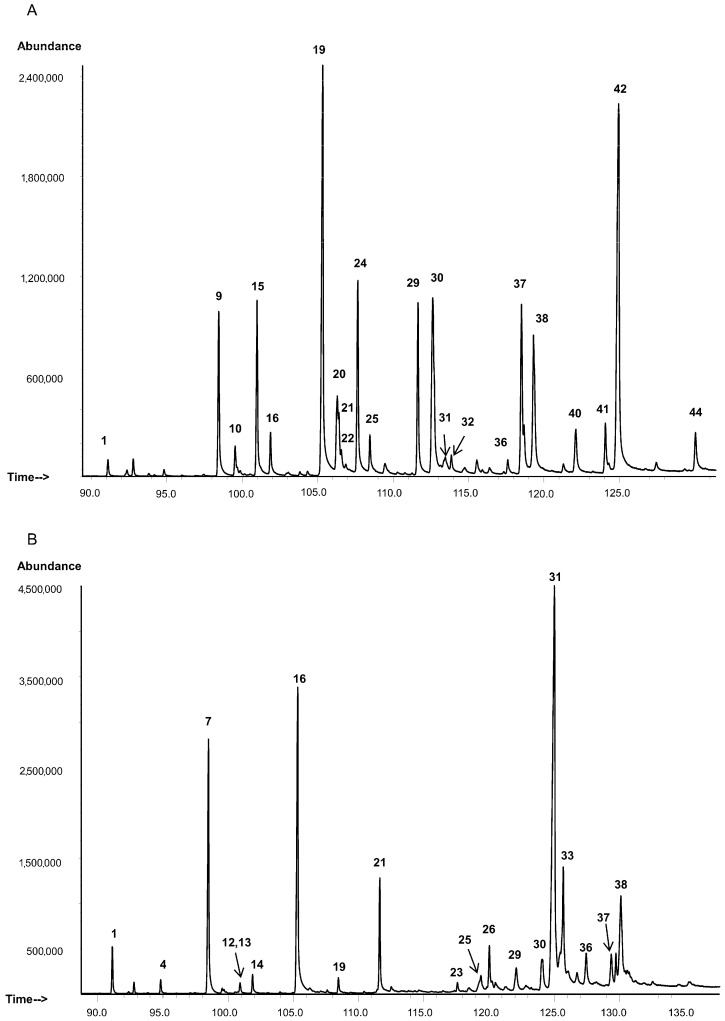
The chromatogram of the hydrocarbon profile of (**A**) *L. migratoria* and (**B**) *C. italicus*. Peak numbers correspond to Appendix A. Only peaks subjected to quantification are presented (area > 0.5%). Arrows indicate peaks belonging to certain numbers. Minor peaks are given in Appendix A.

**Figure 2 insects-13-00736-f002:**
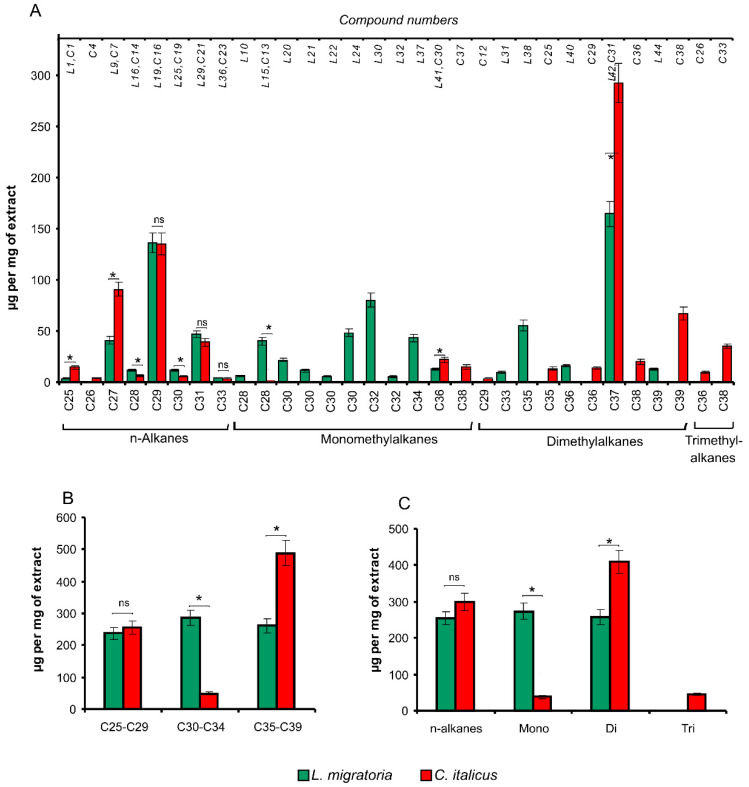
Hydrocarbon composition of epicuticular extracts from *L. migratoria* and *C. italicus* nymphs. (**A**)**.** The distribution of individual hydrocarbons. The lower *x*-axis indicates the total carbon number, and the upper *x*-axis presents numbers of compounds in *L. migratoria* (*L*) and *C. italicus* (*C*) corresponding to Appendix A. (**B**). The carbon number distribution. (**C**)**.** The distribution of methyl branches. * Significant differences between the species (*t*-test, *p* < 0.05, or Mann–Whitney *U* test, *p* < 0.05); ns: nonsignificant differences.

**Figure 3 insects-13-00736-f003:**
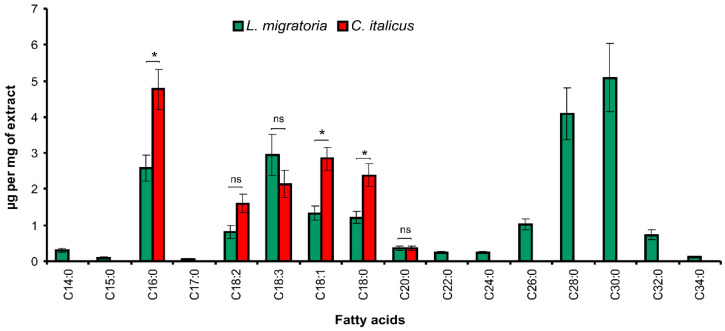
The profile of fatty acids in epicuticular extracts of *L. migratoria* and *C. italicus* nymphs. * Significant differences between the species (*t*-test, *p* < 0.05); ns: nonsignificant differences.

**Figure 4 insects-13-00736-f004:**
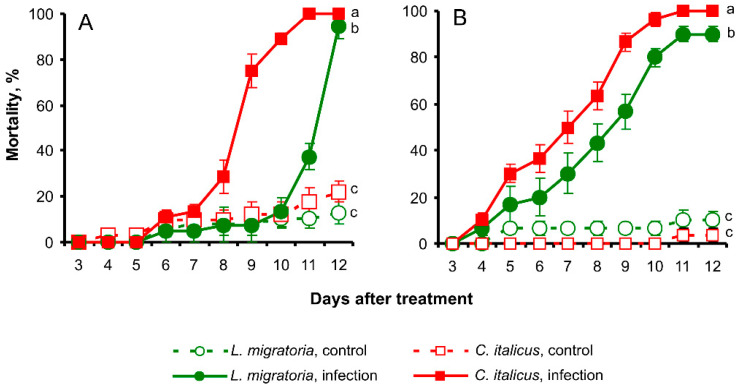
Mortality dynamics of *L. migratoria* and *C. italicus* nymphs after immersion in an *M. robertsii* (**A**) or *B. bassiana* (**B**) suspension (10^7^ conidia/mL). Different letters indicate significant differences in mortality dynamics (log-rank test: χ^2^ > 6.8, df = 1, *p* < 0.009).

**Figure 5 insects-13-00736-f005:**
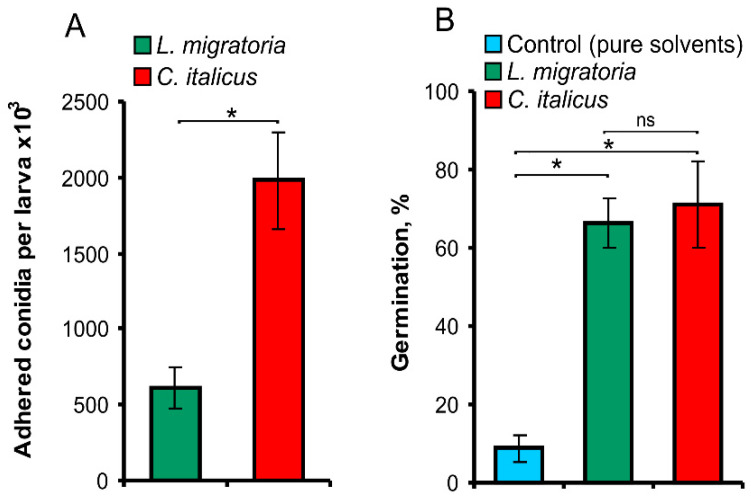
The number of *M. robertsii* conidia adhering to *L. migratoria* and *C. italicus* cuticles after dipping in a suspension containing 10^8^ conidia/mL (**A**), and conidia germination on the *L. migratoria* and *C. italicus* epicuticular extracts applied to the agarose surface (**B**). * Significant differences between the species (*t*-test, *p* < 0.003); ns: nonsignificant differences.

## Data Availability

The data presented in this study are available in the article and Appendix A.

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
