# Peer review of "Comparative Analysis of Epicuticular Lipids in Locusta migratoria and Calliptamus italicus: A Possible Role in Susceptibility to Entomopathogenic Fungi"

_insects, 2022, doi:10.3390/insects13080736_

Round 1

Author Response

 Comment. The manuscript is generally well written; there are only a few typos which need correction, for example, Line 85 habits should be habitats; line 475 susceptibility should be susceptible.

Response. We are grateful for work with our paper and valuable comments. L85 and L475 corrected as required.

Comment. There is inappropriate terminology used throughout: larvae should be corrected by nymphs (or less formal hoppers) as this is an accepted term for hemimetabolous insects and in particular, in acridology.

Response. Corrected as required.

Comment. The topic of the study is novel and the experimental design appears appropriate. However, there are some questionable assumptions. On Line 87, it is stated that “S. gregaria… possesses hygrothermal preferences similar to those of the Italian locust.” This is not true. S. gregaria is truly a species from [mostly] subtropical dry regions while C. italicus is a temperate species inhabiting steppes. So, both in terms of temperature and precipitation requirements these two species are very different and therefore any ecological parallels between S. gregaria and C. italicus are questionable to say the least. As such, they should be avoided and withdrawn from the article.

More than that, certain authors speculate that the common name “Desert Locust” for S. gregaria is not appropriate because in fact, this species needs at least 25 mm of rain to lay eggs and thus, it should be considered a mesophilous insect exploiting temporary humid habitats in an overall dry milieu (see, for example, Duranton & Lecoq, 1990 – Le Criquet pèlerin au Sahel). If this is accepted, the Desert Locust has ecological requirements which are closer to the Migratory, but not to Italian Locust.

Response. We agree. Thank you for pointing out an important mistake. All comparisons in ecology of these two species (S. gregaria and C. italicus)  have been removed from the text (Please see revised version). Only comparisons of cuticle biochemistry and susceptibility to fungi were left. Text in discussion regarding S. gregaria, L. migratoria and C. italicus ecology was rewritten and supplemented. Now it look as:

«Those authors [17] proposed that the large n-alkane amount in S. gregaria serves as an adaptation to the more arid climate, and di- and trimethylalkanes in the S. gregaria epicuticle expand the melting point range and provide stability to the lipid matrix during wide daily fluctuations of temperature in arid habitats. Nonetheless, from the ecological point of view, S. gregaria exploits temporary humid habitats in a milieu that is dry overall [47]. It has been shown experimentally that the optimal humidity range is similar between S. gregaria and L. migratoria migratoriodes (60–70%) [48-49]. Moreover, the ranges of temperature and humidity in which S. gregaria hoppers can develop are narrower as compared to L. migratoria migratoriodes hoppers. In particular, L. migratoria migratoriodes hoppers are better adapted to high humidity (80%) [48]. Therefore, the question of the correlation between cuticular hydrocarbon composition and adaptations to habitats within this pair of species remains open. To our knowledge, there are no comparative studies on thermo- and hygropreferences of C. italicus and L. migratoria migratoria. Nevertheless, it is known that in the studied region, the Italian locust exploits a wider range of habitats and prefers semideserts and dry steppes at different altitudes, whereas the migratory locust shows narrower ecological requirements and prefers reed beds along water bodies [21,22,50,51]. It is possible that the shifts to dimethylalkanes and trimethylalkanes in C. italicus (as compared to L. migratoria) are an adaptation to a wide range hygrothermal conditions and/or their fluctuations in arid landscapes.»

Added References:

  1. Duranton, J.-F.; Lecoq, M. Le Criquet pèlerin au Sahel. CIRAD-PRIFAS, Montpellier, 1990.
  2. Hamilton, A.G. The relation of humidity and temperature to the development of three species of african locusts—Locusta migratoria migratorioides (R. & F.), Schistocerca gregaria (FORSK.), Nomadacris septemfasciata (SERV.). Trans. R. Ent. Soc. Lond. 1936, 85, 1-60. https://doi.org/10.1111/j.1365-2311.1936.tb00231.x
  3. Hamilton, A.G. Further studies on the relation of humidity and temperature to the development of two species of african locusts –Locusta migratoria migratorioides (R. & F.) and Schistocerca gregaria (forsk.). Trans. R. Ent. Soc. Lond. 1950, 101, 1-58. https://doi.org/10.1111/j.1365-2311.1950.tb00374.x
  4. Sergeev, M.G.; Childebaev, M.K.; Van'kova, I.A.; Gapparov, F.A.; Kambulin V.E.; Kokanova, E.O.; Lachininskiy, A.V.; Pshenitsina, L.B.; Temreshev, I.I.; Cernyakhovskiy, M.E.; Sobolev, N.N.; Molodtsov, V.V. Italian locust Calliptamus italicus (Linnaeus, 1758). Morphology, ecology, distribution, menegment of populations. FAO, Rome, 2022 https://doi.org/10.4060/cb7921ru (In Russian).
  5. Latchininsky, A.V.; Sergeev, M.G.; Childebaev, M.K.; et al. Acridids of Kazakhstan, Central Asia, and the Adjacent Territories. Association for Applied Acridology International, Wyoming University, Laramie, 2002 (in Russian).

Comment. In order to have a meaningful comparison of different species in terms of hygrothermal requirements, the authors should have compared the Migratory Locust with a Moroccan Locust Dociostaurus maroccanus, which is truly a xerophilous species. The authors stated that Migratory Locust is less susceptible to fungal infection, but maybe this is because it inhabits areas with much higher temperatures than C. italicus, and such temperatures inhibit fungal growth? Another question is the natural availability of Metarhizium fungus: maybe, it is simply not as common in sandy soils near reed beds where Migratory Locust lives, than in steppic soils?

Response. Steppe complex of Dociostaurini species was more susceptible to fungi compared to L. migratoria. Dociostaurini susceptibility is more close to C. italicus in laboratory condition (it is indicated in discussion). Unfortunately, there are no studies on lipid composition of Dociostaurini. Regarding soil, we agree that is very interesting question. Undoubtedly sandy soils is less appropriate for fungal persistence (Jaronski et al., 2008). However in present work we analyze susceptibility only in laboratory condition. Fungal load in natural habitats of locusts should be separate study. Notably in studied region, temperature in C. italicus habitats (dry steppes and semi-deserts) is not much higher compared to L. migratoria habitats (M.G. Sergeev, pers. comm.). Moreover high UV radiation and low humidity are crucial for survival and development of fungi on terrestrial stages of insect. In conditions of C. italicus habitats (sparse grass and more high altitudes), UV values are higher and humidity is lower compared to L. migratoria habitats (reed beds along water bodies). Under high UV and low humidity, it is much more difficult for fungi to survive and develop on the cuticle. We have added a few sentences regarding this in penult paragraph of the Discussion and Conclusion sections. Please see revised version.

Comment. I also found questionable the interpretation of Figure 4. Based on a shorter LT50, the authors concluded that C. italicus is more susceptible to Metarhizium infection than L. migratoria. However, at day 12, the differences in mortality of the two species are insignificant. This means that both locust species are equally susceptible to the fungal infection – which contradicts the authors’ conclusions. Big (although statistically insignificant) difference in mortality of control populations of C. italicus and L. migratoria (21 and 13%) is surprising and I could not understand if it was taken into account when mortalities of the treated cohorts were calculated.

Response. We added information on Abbott’s adjustment in M&M and Results sections. The effect remained at a high level of significance (P < 0.015 from day 7 to day 11). Regarding LT50, as a rule LT50 and LC50 are strongly correlated. Unfortunately we did not have a large number of hoppers to test many concentrations. However LT50 extensively used as parameter of susceptibility to fungal infection even if the mortality reaches 90-100% (e.g., doi.10.1073/pnas.1703546114,  doi.10.1073/pnas.1616543114).

Comment. Finally, I am concerned with the authors’ overstatement: they studied the infection of just one isolate of one fungus species, M. robertsii, but they refer to it as “fungal infection.” Unless several fungal species are studied, I suggest to use the term “fungal infection by an isolate of M. robertsii” to avoid the incorrect generalization.

Response. Thank you for valuable comment. We supplemented MS by additional data on bioassays with B. bassiana and added one co-authors Georgy Lednev conducted these experiments on same populations. There is same effect after treatment with B. bassiana. Correspondent supplementations were added in M&M, Results, Discussion and Conclusion sections. Please, see revised version.  

We would be to grateful for further constructive comments.

Reviewer 2 Report

1. Introduction, L41, the author described the functions and importance of insect surface lipids, Does the surface and cellular lipid possess any difference in structure, storage, and metabolism? if so please explain in the background section.

2. There are several reports about the advancement of CHCs biosynthesis, How the biosynthesis of hydrocarbons happens? how is the transportation process, and which proteins are involved in transporting the hydrocarbons to the cuticle? The author should add this information. 

3. lipid homeostasis is important for a healthy individual and its dysregulation causes many metabolic problems, I would suggest the author briefly explain lipid storage, lipogenesis, oxidation, and mobilization. How in locust do these processes take place and what are the consequences of dysregulation?

4. L87-89, Please rewrite these sentences.

5. L93, Please rewrite the sentence.

6. L115, Please rewrite the sentence.

7. L121, why the analysis was performed only in the larval stage of the locust? 

8. L221, Please report the statistics

9. In my opinion, please start every heading in the results with a background sentence, states that what is the rationale and importance of this analysis?

10. Figure 5B, are these results non-significant? If so, then here and everywhere in MS, please indicate non-significant results with (ns) and explain in the legends.

11. In the discussion section, the author should add information that how the CHCs profiling which varies with habitat, has the potential to control the pest? what control strategies would adopt to interfere with CHCs? 

12. In the conclusion section, 503-505, please rewrite the sentence with the correct wording. 

13. There is no information about the adult stage of locust, please if possible explain which stage of the pest is destructive? also, during pupal stages many changes happen, do you have any understanding that the larval CHCs would vary or will have the same profile in the adult stage? 

Author Response

Comments 1 and 2. Introduction, L41, the author described the functions and importance of insect surface lipids, Does the surface and cellular lipid possess any difference in structure, storage, and metabolism? if so please explain in the background section.    There are several reports about the advancement of CHCs biosynthesis, How the biosynthesis of hydrocarbons happens? how is the transportation process, and which proteins are involved in transporting the hydrocarbons to the cuticle? The author should add this information

Response. Thank you for work with our paper and valuable comments. Information was added in Introduction section:

«Cuticular lipid composition is significantly different from that of hemolymph and fat body, which contain mainly triglycerides and diglycerides [7]. Insect cuticular hydrocarbons are synthesized in oenocytes of the epidermal layer of integuments and are transported by lipophorin to epidermal cells via pore canals and epicuticular channels [6]. Both linear and branched hydrocarbons may be synthesized in different insect taxa, and mechanisms of their synthesis have been reviewed by Blomquist [8]. Briefly, hydrocarbons are synthesized from long-chain fatty acetyl-coenzyme A by reductive decarboxylation. The elongation of the carbon chain proceeds via the addition of two carbon units by a malonyl-coenzyme A derivative, and the addition of methyl branches occurs by the insertion of propionate (a derivative of methylmalonyl-coenzyme A). The hydrocarbons can be oxidized to secondary alcohols by cytochrome P450 enzymes and oxygen (O2) and then can be oxidized to ketones and form wax esters [9].»

Added References:

  1. Lockey, K.H. Lipids of the insect cuticle: origin, composition and function. Comp. Biochem. Physiol. B: Comp. Biochem., 1988, 89, 595-645. https://doi.org/10.1016/0305-0491(88)90305-7
  2. Downer, R.G.H.; Matthews, J.R. Patterns of Lipid Distribution and Utilisation in Insects. American Zoologist, 1976, 16, 733–745. https://doi.org/ 10.1093/ICB/16.4.733
  3. Blomquist, G. Biosynthesis of cuticular hydrocarbons. In Insect Hydrocarbons Biology, Biochemistry, and Chemical Ecology, 1st ed.; Blomquist, G. J., Bagnères, A.-G., Eds.; Cambridge University Press: New York, NY, USA, 2010; pp. 35-52. https://doi.org/10.1017/CBO9780511711909.004
  4. Buckner, J. Oxygenated derivatives of hydrocarbons. In Insect Hydrocarbons Biology, Biochemistry, and Chemical Ecology, 1st ed.; Blomquist, G. J., Bagnères, A.-G., Eds.; Cambridge University Press: New York, NY, USA, 2010; pp. 187-204. https://doi.org/10.1017/CBO9780511711909.010

Comment. 3. lipid homeostasis is important for a healthy individual and its dysregulation causes many metabolic problems, I would suggest the author briefly explain lipid storage, lipogenesis, oxidation, and mobilization. How in locust do these processes take place and what are the consequences of dysregulation?

Response. Unfortunately, we cannot fully agree with this suggestion. Mechanisms of lipid storage, lipogenesis, oxidation, and mobilization are described in detail in works of Blomquist and coworkers whom we cite (please, see previous response).  Our paper is dedicated only to epicuticular lipids in terms of adaptation to abiotic factors and fungal pathogeneses, but not to lipid biosynthesis and mobilization. Please, let us do not include this information because it defocus the reader and overload the introduction. Regarding lipid dysregulations, we supplemented introduction with point of view of fungal pathogeneses:

«Hydrocarbon composition and fatty acid composition of the insect cuticle and of internal tissues significantly change during the development of a fungal infection; however, these effects are not straightforward and depend on insect and pathogen species [32-35]. Probably, the alterations are caused not only by direct utilization of lipids by fungi but also by changes in host metabolic pathways during fungal infections. Notably, a number of fatty acids (especially with short chains) can inhibit germination and mycelial growth of entomopathogenic fungi in vitro [36,37] and can influence the activity of virulence-related enzymes [38,39].»

 Added References:

  1. Lecuona, R.; Riba, G.; Cassier, P.; Clement, J.L. Alterations of insect epicuticular hydrocarbons during infection with Beauveria bassiana or B. brongniartii. J. Invertebr. Pathol. 1991, 58, 10-18, https://doi.org/10.1016/0022-2011(91)90156-K.
  2. Gołębiowski, M.; Cerkowniak, M.; Ostachowska, A.; Naczk, A.M.; Boguś, M.I.; Stepnowski, P. Effect of Conidiobolus coronatus on the cuticular and internal lipid composition of Tettigonia viridissima males. Chem. Biodiversity, 2016, 13, 982-989. https://doi.org/10.1002/cbdv.201500316
  3. Paszkiewicz, M.; Gołębiowski, M.; Sychowska, J.; BoguÅ›, M.I.; WÅ‚óka, E.; Stepnowski, P. The effect of the entomopathogenic fungus Conidiobolus coronatus on the composition of cuticular and internal lipids of Blatta orientalis females. Physiol. Entomol. 2016, 41, 111-120. https://doi.org/10.1111/phen.12133
  4. Gołębiowski, M; Bojke, A; Tkaczuk, C Effects of the entomopathogenic fungi Metarhizium robertsii, Metarhizium flavoviride, and Isaria fumosorosea on the lipid composition of Galleria mellonella larvae. Mycologia, 2021, 113, 525-535. https://doi.org/10.1080/00275514.2021.1877520

Comment. 4. L87-89, Please rewrite these sentences.

Response. Corrected: “There are studies describing cuticular lipids of adult L. migratoria ssp. migratorioides [17,23,24], but there are no studies about cuticular lipids of C. italicus. Nevertheless, there are research articles describing cuticular lipids of S. gregaria, which inhabits subtropical dry regions.”

Comment. 5. L93, Please rewrite the sentence.

Response. Corrected: “Those authors proposed that these properties of the S. gregaria lipid layer may represent an adaptation to arid habitats because a large n-alkane amount decreases desiccation.”

Comment. 6. L115, Please rewrite the sentence.

Response. Corrected: “To the best of our knowledge, there are no research papers describing the impact of epicuticular lipids of different locust species on their susceptibility to entomopathogenic fungi.”

Comment. 7. L121, why the analysis was performed only in the larval stage of the locust? 

Response. Experimentation on nymphs allows us to accurately link their origin to a certain habitats, while adults can migrate significantly. In addition, the use of nymphs allows us to level the time after molting, since they have a shorter development period. The adult lives for a long time and can be caught at very different times after the molt. We have indicated this in the first paragraph of the Materials and Methods section. Please, see revised MS.

Comment. 8. L221, Please report the statistics

Response. Information added, differences are nonsignificant “(t = 0.52, df = 10, P = 0.62)”

Comment. 9. In my opinion, please start every heading in the results with a background sentence, states that what is the rationale and importance of this analysis?

Response.  Done:

3.1 Epicuticular hydrocarbons, fatty acids, and other lipids of C. italicus and L. migratoria were analyzed to understand their possible involvement in adaptation to different habitats and in susceptibility to fungal infections.

3.2. Comparative analysis of mortality dynamics of C. italicus and L. migratoria nymphs was performed after treatment with M. robertsii and B. bassiana conidia.

3.3. Prepenetration stages of fungal development on the cuticle (adhesion and germination) are dependent on the host’s lipid composition, which may determine the susceptibility of insects to fungal pathogens [3,27,28]. Therefore, we estimated the number of M. robertsii conidia adhering to the cuticle of both locust species in vivo and the rate of conidia germination in vitro.

Comment. 10. Figure 5B, are these results non-significant? If so, then here and everywhere in MS, please indicate non-significant results with (ns) and explain in the legends.

Response. Done for all figures and legends. Please see revised MS

Comment. 11. In the discussion section, the author should add information that how the CHCs profiling which varies with habitat, has the potential to control the pest? what control strategies would adopt to interfere with CHCs? 

Response. We are sure that it is too early investigation for these conclusions. The work does not concern effects in the wild, but only in the laboratory. Humidity and UV radiation will play a key role in the field. There is a series of works on the change in fungal virulence when growing fungi on hydrocarbons and fatty acids, but this is out of frames of our study. We have only indicated in the Discussion and Conclusion that field studies of infection load and susceptibility of locust in different habitats will be promising. Please see revised version.

Comment. 12. In the conclusion section, 503-505, please rewrite the sentence with the correct wording. 

Response. Conclusion was corrected and looks as:

This is the first comparative analysis of cuticular lipids of L. migratoria and C. italicus nymphs. Linear and methyl-branched alkanes are predominant in the lipid profiles of both species, although the lipid profiles are considerably different between these species: 1) C. italicus contains a larger amount of long-chain hydrocarbons (С35–С39) as compared to L. migratoria; 2) C. italicus possesses a lower amount of monomethylalkanes but a larger amount of di- and trimethylalkanes relative to L. migratoria; 3) n-alkane proportion is slightly higher in C. italicus; and 4) C. italicus has a narrower spectrum of fatty acids than L. migratoria does. Higher susceptibility to fungal infections caused by M. robertsii and B. bassiana were registered for C. italicus compared to L. migratoria. The greater number of M. robertsii conidia adhering to the cuticle was observed, which correlated with an elevated amount of long-chain hydrocarbons in C. italicus. We believe that epicuticular hydrocarbons of C. italicus may be an adaptation to a wide range of habitats (including arid steppes and semideserts) but may make the C. italicus cuticle more hospitable for fungi. Because a propensity for higher susceptibility to fungi has been documented for other xerophilous locusts compared to mesophilic ones in laboratory assays, future research may be devoted to an immune response to fungi in locusts with distinct hygrothermal preferences as well as to the development of fungal infections in natural habitats.

Comment. 13. There is no information about the adult stage of locust, please if possible explain which stage of the pest is destructive? also, during pupal stages many changes happen, do you have any understanding that the larval CHCs would vary or will have the same profile in the adult stage?

Response. We added the comparisons between locust adults and nymphs in discussion: “Notably, the hydrocarbon profile of L. migratoria migratoria nymphs established in the present paper is very similar to the profile of L. migratoria migratoriodes adults described by Lockey and Oraha [17]. Despite differences in developmental stages and subspecies, the same chain lengths for different classes of alkanes (n-alkanes, monomethylalkanes, and dimethylalkanes) are predominant, and their relative abundance levels are similar too»

Regarding destructivity, both adults and nymphs may be destructive, but treatment of nymphs is more effective in both biological and chemical controls, because the larvae are localized in limited territories and the adults may fly hundreds of kilometers. However it does not appear that at this stage of research our data can contribute to locust biocontrol.

We would be to grateful for further constructive comments

Round 2

Reviewer 1 Report

The authors sufficiently addressed the reviewers' comments and made appropriate changes.

Correct the spelling in Ref. 50: should be Latchininsky

Author Response

Comment. The authors sufficiently addressed the reviewers' comments and made appropriate changes. Correct the spelling in Ref. 50: should be Latchininsky

Response. Thank you for work with paper. Corrected as requirerd. Some minor mistakes in transliteration and translation in these REFs was also corrected as recomended by one of author of the books Mikhail Sergeev